# Ethnobotanical Research on Dye Plants Used by the Baiyi Indigenous Peoples' from Heqing County, Dali, Yunnan, China

Rong Yang [1,2], Shengji Pei [1], Yuying Xie [3], Xiuxiang Yan [2], Angkhana Inta [2,*] and Lixin Yang [1,4,*]

1 Key Laboratory of Economic Plants and Biotechnology, Kunming Institute of Botany, Chinese Academy of Sciences, Kunming 650201, China; yangrong@mail.kib.ac.cn (R.Y.); peishengji@mail.kib.ac.cn (S.P.)
2 Department of Biology, Faculty of Science, Chiang Mai University, Chiang Mai 50200, Thailand; yanxiuxiang@mail.kib.ac.cn
3 School of Agriculture and Food Sciences, University of Queensland, Brisbane, QLD 4072, Australia; yuying.xie@uq.net.au
4 Center for Biodiversity and Indigenous Knowledge, Kunming 650034, China
* Correspondence: aungkanainta@hotmail.com (A.I.); rattan@mail.kib.ac.cn (L.Y.)

**Abstract:** With the advantage of being eco-friendly, plant dyes have been noticed by textile practitioners and the public. However, as a result of the rapid advancements in industrial manufacturing, the traditional knowledge of plant dyes is dying, which demands heightened attention and protection. To document this traditional knowledge in the Baiyi community, semi-structured interviews were conducted with 288 informants from four villages (Five stars village, Hedong village, Nanpo village, and Shang' eping village) of the Liuhe Yi Nationality Township, Heqing County, Dali Prefecture. Based on the ethnobotanical investigation of plant dyes, there were 11 plant species from 11 genera in 10 families that have been used as dyes. The Baiyi Indigenous peoples mainly extract the dye from the roots, leaves, fruits, etc., of the herbaceous plants. Through quantitative analysis, the frequency of use (f value) and cultural importance indices (CI value) of *Viburnum cylindricum* are the highest. The optimal conditions for dyeing cotton fabric with *V. cylindricum* were found to be a pH of 5, dyeing for 30 min at 60 °C by an orthogonal array design. As for the dyeing properties, biomordants provided better properties when used in dyeing cotton fabric with *V. cylindricum* compared with metallic mordants. This study reveals the great potential of the application of plant dyes in the Baiyi Indigenous peoples community; it will be beneficial to the economic development of ethnic areas, the inheritance of ethnic culture and the protection of biodiversity.

**Keywords:** ethnobotany; plant dye; Baiyi indigenous peoples'; traditional knowledge; craft optimization





## 1. Introduction

Dyes contribute to many aspects of human life. Natural dyes, which can be obtained from plants, animals, and microorganisms, have a long history of utilization [1,2]. Plant dyes are normally extracted from roots, stems, leaves, flowers, fruits, and other parts of the plant, so they are considered to be the most convenient and widely used in natural dyes [3]. Under the development of industrialization, natural dyes have been gradually replaced by synthetic dyes with the advantages of low cost, convenience and better dyeing properties [4]. However, the disadvantages of synthetic dyes have come into focus. Synthetic dyes can cause environmental pollution and be harmful to human health [5,6]. For example, azo dyes are carcinogenic [7], and the heavy metals in the wastewater produced during synthetic dyeing are toxic to the environment [8,9].

Plant dyes have many advantages including being renewable, biodegradable, and eco-friendly, which is drawing more attention from researchers, textile practitioners and the public [10–12]. In addition, many plant dyes have medicinal and health benefits for the human body. For example, turmeric has antioxidant, antitumor, anti-inflammatory, neuroprotective, hepatoprotective, and cardioprotective effects [13]; *Baphicacanthus cusia*

(Nees) Bremek. can clear heat, detoxify the blood, eliminate spots and help alleviate anxiety [14]. Research on plant dyes is normally focused on their utilization in producing functional textiles, cosmetics, food, and solar cells, which suggests the bright future of plant dyes [15–17].

On the other hand, there are also disadvantages of plant dyes including low color fastness, limited color range, limited availability, and high costs [11,18,19], which limit the industrial application of plant dyes. Mordants, especially metallic mordants, can improve the dyeing properties of plant dyes [20,21]. Although metallic mordants can enhance the color fastness of plant dyes, they can pollute the environment, damage human health, and destroy the ecosystem [22,23]. By contrast, biomordants enhance the dyeing properties of plant dyes and are biodegradable and not harmful to human health [24,25]. For instance, gum rosin, soybean, and mud are used as biomordants by Chinese minorities to improve plant dyes [26,27]. To make plant dyes a viable alternative to synthetic dyes, it is necessary to discover more sources of plant dyes and develop plant dyeing technology to improve their dyeing properties.

Worldwide remote indigenous communities have developed traditional dyeing techniques using local species of plants [28–30]. Ethnic communities often use plants to dye food, clothing, handicrafts, and fingernails [31,32]. The diversity of dye plants and traditional plant dyeing knowledge can provide scientific clues for their research and development [27]. Ethnobotanical studies of plant dyes and the relevant traditional knowledge have been conducted in indigenous communities worldwide [28,29,33,34]. Current research has noticed that plant dyeing and related traditional knowledge is gradually disappearing because of the industrial modernization and impact of modern popular culture [28,32]. The diversity of ethnic cultures, including dyeing culture, can provide cultural resources and dynamic support for sustainable development. Thus, ethnic cultures should be respected and protected [33,35]. There is an urgent need to collect, organize, and record ethnic knowledge and technology of plant dyeing.

Ethnic minorities still use traditional plant dyeing crafts and have rich plant dyes resources in Dali, Yunnan Province, China. Meanwhile, there are twenty-three species of plant dyes [31]. The Baiyi Indigenous peoples, a branch of the Yi ethnic group in Heqing, Dali China, have the custom of traditional plant dyeing [36]. Most Baiyi Indigenous peoples (who call themselves "Kuaeshi") live in Liuhe Yi Nationality Township in Heqing County, Dali. There is a legend that the ancestors of the Baiyi Indigenous peoples migrated from Lianhuataru. Although the Baiyi Indigenous peoples have their own language, it cannot be written, so their traditional culture can only be passed to the next generation orally [37]. The traditional ethnic culture of the Baiyi Indigenous peoples is closely related to the local natural environment, which is represented by the traditional Huo Cao Yi clothing. The Baiyi Indigenous peoples use dye plants and plant fibers to make Huo Cao Yi cloth [37]. Therefore, traditional knowledge systems related to plant dyeing have been generated by the Baiyi Indigenous peoples and passed on over generations. In years gone by, the Baiyi Indigenous peoples were geographically and linguistically isolated, which allowed the original ecological preservation of the ethnic culture. However, only a few studies have focused on plant dyeing and the related traditional knowledge of the Baiyi Indigenous peoples.

The purpose of this research is to preserve plant dyeing and related traditional knowledge in the Baiyi community, Liuhe Yi Township, Heqing County, Dali Prefecture, China, optimize traditional plant dyeing techniques and lay the foundation for developing and using plant dye resources in sustainable industrial production. Thus, the three aspects are the focus of this study: (1) The current situation of resources of plant dyes in Heqing County; (2) the traditional plant dyeing craft and related traditional knowledge of the Baiyi community; (3) the optimized dyeing conditions of traditional plant dyeing.

## 2. Materials and Methods

### 2.1. Study Area

This study was conducted in Liuhe Yi Nationality Township, which is the only ethnic township in Heqing County, Dali Bai Autonomous Prefecture, Yunnan Province, China (Figure 1). Liuhe Yi Nationality Township is located in southeastern Heqing County. The altitude ranges from 1680 to 2748 m. The township covers an area of 249.5 km², with 13 administrative villages. It has a subtropical monsoon climate with no severe cold in the winter and no extreme heat in the summer. The annual average temperature is 14.7 °C, and the annual rainfall is ~900 mm [36]. Bai, Yi, Han, Miao, and other ethnic groups live in the area. The Baiyi Indigenous peoples, comprising one of the branches of the Yi nationality, live in the area, which is the earliest birthplace of Baiyi culture in Yunnan Province [37]. The Baiyi Indigenous peoples, who call themselves the "Kuaeshi family," have developed a traditional knowledge system of plant dyeing during long production and life.

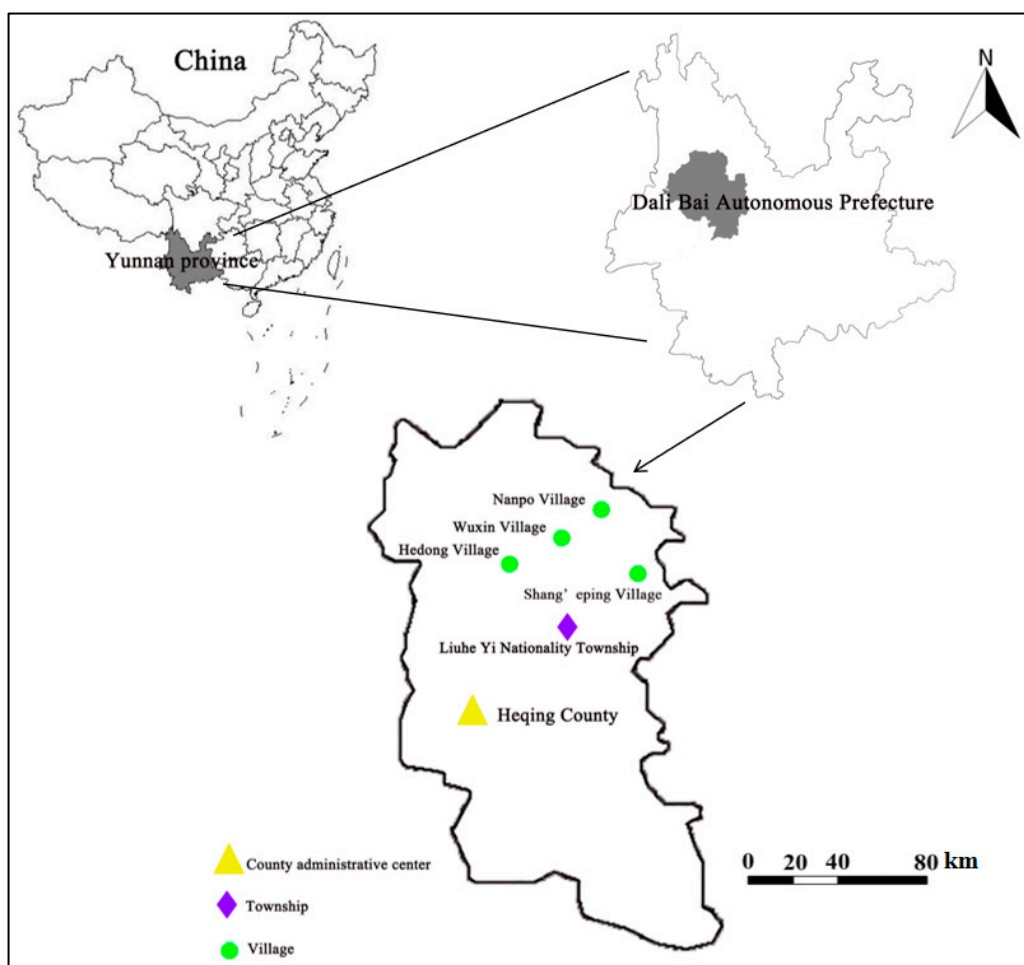

**Figure 1.** Geographical distribution map of the study sites.

The preservation degree of the traditional plant dyeing craft of the Baiyi Indigenous peoples was used as the selection criterion for the study site. Based on field investigations and the relevant literature [37], Five stars village, Hedong village, Nanpo village, and Shang'eping village of Liuhe Yi Nationality Township were selected as the study areas (Figure 1). Most Baiyi Indigenous peoples still use their ethnic language to communicate, and the traditional Baiyi culture has remained intact in these villages.

## 2.2. Field Research

Three ethnobotanical surveys on traditional plant dyeing were conducted in four villages of Liuhe Yi Nationality Township from September 2019 to March 2020. The interview mode of 5W + 1H [38,39] was employed to conduct semi-structured interviews with key informants ("inheritors" and "old people") and relevant institutions (the local village committee and Baiyi Cultural Inheritance Park). A total of 288 informants (158 women and 130 men, ages 35–85) were interviewed in the local community. The questions asked in the interviews are shown in Table 1. The snowball sampling method and questionnaire were adopted to collect original data [40,41]. All informants gave verbal informed consent prior to the interview [42]. After obtaining permission, we took photographs of the traditional plant dyeing craft.

**Table 1.** Outline of a semi-structured interview.

| Question |
| --- |
| 1. Which major plant species do you use for traditional dyeing? |
| 2. What is the local name of this plant dye? |
| 3. What color dye does this plant produce? |
| 4. Which plant part (root, stem, leaf, flower, fruit, peel, whole plant, seed, bark) is utilized? |
| 5. Which methods are often used to extract plant dye? |
| 6. What time of year is the plant dye usually collected? |
| 7. How do you use this plant dye? |
| 8. Do you need to add anything else to the plant dye for traditional dyeing? |
| 9. What are the effects of the additive on the dyeing process? Is it added before or after dyeing? Why? |
| 10. Where can we collect this plant dye? |
| 11. What other uses does this plant species have besides being used to make dye? |
| 12. Who collects plant dyes? |
| 13. Who extracts the dyes from the plants? |
| 14. Who performs the dyeing using the extract? |
| 15. What is the story behind plant dyes? |
| 16. How are the crafts of traditional plant dyeing passed down to the next generation? |
| 17. Is the plant dye used to dye cloth, food, for decoration, or anything else? |
| 18. Is this plant dye still used? If not, why? |
| 19. Is this plant dye used only on certain festivals or days? |
| 20. Who taught you to dye with plant dyes? When did you learn to dye? |

The local plant dyes were identified, classified and collected as document specimens and samples to form the ethnobotanical catalog of plant dyes of the Baiyi Indigenous peoples. The scientific names of dye plants were classified and confirmed through APG IV. And plant specimens were reserved at the Herbarium, Kunming Institute of Botany, CAS. Moreover, we recorded the traditional plant dyeing craft of the Baiyi Indigenous peoples via participatory observation.

## 2.3. Data Analysis

The use frequency (f) and cultural significance (CI) were used as indicators to assess the importance of the plant dye resources and related traditional knowledge in the Baiyi community by quantitative analysis [31].

The use frequency (f) was assessed based on the usage degree of each dye plant, which was calculated using the following formula [31]:

$$f = N_m/N_i \tag{1}$$

where f is the use frequency of each plant dye, $N_m$ is the frequency of certain plant species mentioned by informants, and $N_i$ is the sum of the number of informants. The higher the f value, the more frequently the plant dye is utilized in the local community and the higher the potential value for development.

The cultural significance (CI) value was used to evaluate the usage diversity and recognition degree of each plant dye in the local community following Tardío and Pardo-de-Santayana [27], which was calculated using the following formula:

$$CI_S = \sum_{u=u_1}^{uNC} \sum_{i=i_1}^{iN} \frac{UR_{ui}}{N} \tag{2}$$

where N is the total number of informants, NC is the total number of applications for the species of dye plant S, and $UR_{ui}$ is the utilization report (UR) for the plant species S which is described by the informant for application u. Therefore, the CI is the total proportion of informants who referred to each type of use of a plant species.

The index reflects the diversity of utilization of each plant species and the awareness degree of the application of certain plant species by informants. A high CI value indicates that the plant has a wide application value, and it is well-known by most people.

### 2.4. Materials

*Viburnum cylindricum* Buch.-Ham. ex D. Don and black mud were collected in October 2019 from Wuxing Village (26°41′93′′ N, 100°34′13′′ E) of Liuhe Yi Nationality Township, Heqing County, Dali Bai Autonomous Prefecture, Yunnan Province, China. Cotton fabrics were purchased from Xingwei national craft factory in Weishan County. Gum rosin and soybean were purchased from the local market in Heqing County, Dali Bai Autonomous Prefecture of China. Metallic mordants including alum, stannous chloride, and copper sulfate, as well as NaOH and HCl, were analytical reagent grade.

### 2.5. Laboratory Experiments

#### 2.5.1. Extraction of Plant Dye

The fresh leaves of *V. cylindricum* were washed with water to remove impurities; 100 g sample of leaves was placed in 2 L of distilled water and smashed for 10 min to form dye pulp. Then, the pulp was filtered, and the filtrate was diluted with water to 2 L as *V. cylindricum* dye solution [27].

#### 2.5.2. Experimental Design to Optimize the Dyeing Craft

An orthogonal experimental design with temperature (A), time (B), and pH (C) as three factors were used to optimize the dyeing conditions for *V. cylindricum* [26]. Three levels were selected for each factor, as shown in Table 2. The nine dyeing experiments, $L_9$ ($3^3$) of the orthogonal array, are presented in Table 3.

**Table 2.** Factors and levels of the orthogonal array experiment for dyeing fabrics with *Viburnum cylindricum*.

| Level | Factors | | |
|:---:|:---:|:---:|:---:|
| | A. Temperature (°C) | B. Time (Minutes) | C. pH |
| **1** | 40 | 30 | 5 |
| **2** | 60 | 60 | 7 |
| **3** | 80 | 90 | 9 |

#### 2.5.3. Dyeing Crafts

The 2 L *V. cylindricum* dye solution was used to dye cotton (weight 118.7 g m$^{-2}$, plain weave) and hemp (weight 149.1 g m$^{-2}$, plain weave) fabrics at 30, 50, and 70 °C for 40, 60, and 80 min, respectively, with a bath ratio of 30:1. The dyed samples were rinsed with tap water after dyeing and dried at room temperature.

**Table 3.** Orthogonal array design of the nine dyeing experiments based on design L$_9$ (3$^3$).

| No. | A (Temperature) | B (Time) | C (pH) |
|---|---|---|---|
| 1 | A$_1$ | B$_1$ | C$_1$ |
| 2 | A$_1$ | B$_2$ | C$_2$ |
| 3 | A$_1$ | B$_3$ | C$_3$ |
| 4 | A$_2$ | B$_1$ | C$_2$ |
| 5 | A$_2$ | B$_2$ | C$_3$ |
| 6 | A$_2$ | B$_3$ | C$_1$ |
| 7 | A$_3$ | B$_1$ | C$_3$ |
| 8 | A$_3$ | B$_2$ | C$_1$ |
| 9 | A$_3$ | B$_3$ | C$_2$ |

### 2.5.4. Mordanting Methods

The mordanting methods, including pre-, simultaneous, and post-mordanting, were performed at 70 °C for 60 min at a liquor ratio of 30:1 using three metallic mordants (10% of alum, copper sulfate, and stannous chloride) and three biomordants (25 g L$^{-1}$ soybean, 12 g L$^{-1}$ gum rosin, and 300 g black mud).

### 2.5.5. Color Measurements

The lightness (L*), redness-greenness (a*), and blueness-yellowness (b*) of the dyed samples were measured with benchtop spectrophotometer (Shenzhen ThreeNH Technology Co., Ltd., Shenzhen, China) with illuminant D65 and 10° standard observer. Undyed fabrics were used as standard samples. The $K/S$ values of the dyed samples were calculated at a wavelength of maximum absorption (λmax) based on the Kubelka–Munk equation:

$$K/S = (1 - R)\,^2/2R \tag{3}$$

where $R$ is the reflectance of the dyed samples, $K$ is the absorption coefficient, and $S$ is the scattering coefficient of the colorant.

### 2.5.6. Colorfastness Tests

All dyed samples were tested with reference to the Chinese Textiles Test Specification. The color fastness of the dyed samples to washing, perspiration, and rubbing was assessed following the standards GB/T3921-2008, GB/T3922-2013, and GB/T3920-2008, respectively, based on ISO international standards. According to these standards, a scale of 1–5 is used for numerical classification, in which 1 represents poor and 5 represents excellent. In general, level 3 is the basic standard and level 4 or above is receivable for commercial applications.

## 3. Results

### 3.1. Ethnobotanical Study of Dye Plants in the Baiyi Community

#### 3.1.1. The Diversity of Plant Dyes in the Baiyi Community

Based on a field investigation of local plant dyes in the Baiyi community, we generated an ethnobotanical catalog of the dye plants of the Baiyi Indigenous peoples following the identification of evidence specimens (Table 4). It was found that 11 plant species belonging to 10 families and 11 genera are utilized for traditional dyeing by the Baiyi community. The plant species primarily include herbaceous plants (6), followed by shrubs (3), trees (1), and lianas (1). The major plant parts used to extract pigments by the Baiyi Indigenous peoples include roots (3), stems (2), leaves (3), flowers (2), fruits (3), and peels (1), which produce various colors, including green, blue, yellow, white, red, orange, and black. Plant dyes are used as colorants for clothing, foods, and nails in the Baiyi community, especially clothing. Moreover, many kinds of plant dyes have medicinal effects.

**Table 4.** Plant dyes used by the Baiyi communities in Dali Prefecture.

| Specimen Code | Scientific Name | Family Name | Chinese Name | Local Name | Plant Form | Habitat | Plant Part | Dye Color | Usage | Medicinal Uses |
|---|---|---|---|---|---|---|---|---|---|---|
| NP08 | *Buddleja officinalis* Maxim. | Scrophulariaceae | Mi meng hua | Ya wu | Shrub | Sunny limestone slopes | Flower | Yellow | Cloth, food | Clearing heat and expelling "fire," nourishing the liver, and brightening the eyes |
| NP13 | *Camellia sinensis* (L.) O. Kuntze var. assamica (Mast.) | Kitamura | Pu er cha | Pu ai zao se | Tree | The mountains | Leaf | Yellow | Cloth, food | Reducing blood lipids, aiding digestion and weight loss, and warming the stomach |
| SEP08 | *Carthamus tinctorius* L. | Asteraceae | Hong hua | Ho huo | Herb | Sandy soil with good drainage and medium fertility | Flower | Red, yellow, orange | Cloth, food | Easing menstruation, promoting blood circulation, treating gynopathy |
| WX15 | *Curcuma longa* L. | Zingiberaceae | Jiang huang | En ge xi gong | Herb | Woods, meadows, and roadsides | Root | Yellow | Cloth, food | Warming the meridians and promoting blood and gallbladder health |
| NP03 | *Onosma paniculatum* Bur. et Franch. | Boraginaceae | Dian zi cao | Ze tan | Herb | Hillsides, forest margins, and grassy slopes facing the sun | Root | Red | Cloth, nail, food | Clearing heat and cooling the blood, activating the blood, detoxifying, healing rashes, and eliminating spots |
| WX05 | *Oryza sativa* L. | Poaceae | Shui dao | Qie | Herb | The plain | Fruit | White | Cloth, food | Promoting digestion, invigorating the spleen, and increasing the appetite |
| SEP05 | *Punica granatum* Linn. | Lythraceae | Shi liu | Xi mia | Shrub | The mountains | Fruit peel | Yellow | Cloth, food | Hemostasis astringent, repelling and killing insects |
| HD06 | *Rubia cordifolia* L. var. pratensis Maxim. | Rubiaceae | Qian cao | Qian huo huo | Liana | Under open forests, in forest margins, thickets, or grassland | Root, stem | Red | Cloth, nails, food | Cooling the blood to remove blood stasis, hemostasis |
| HD11 | *Strobilanthes cusia* (Nees) Kuntze | Acanthaceae | Ban lan | Na | Herb | Wet underbrush in the woods | Stem, leaf | Blue | Cloth, food | Clearing heat and detoxifying, cooling the blood, and stopping bleeding |

**Table 4.** *Cont.*

| Specimen Code | Scientific Name | Family Name | Chinese Name | Local Name | Plant Form | Habitat | Plant Part | Dye Color | Usage | Medicinal Uses |
|---|---|---|---|---|---|---|---|---|---|---|
| WX01 | *Viburnum cylindricum* Buch.-Ham. ex D. Don | Viburnanceae | Shui hong mu | Xiu xi | Shrub | Open forest or thickets on sunny slopes | Leaf | Green, blue-green, blue, black, brownish black | Cloth, food | Clearing away heat and detoxifying, cleaning abscesses, treating coughs, dysentery, rheumatic pain, and injuries |
| WX09 | *Zea mays* L. | Poaceae | Yu mi | Bao gu | Herb | Sunny slopes | Fruit | White | Cloth, food | Clearing heat and detoxifying, strengthening the spleen and stomach, whitening teeth, and promoting defecation |

Therefore, there are many plant species, plant parts, and applications of plant dyes used by the Baiyi Indigenous peoples (Table 4), indicating that the local resources of plant dye are abundant. Similarly, these findings indicate that the Baiyi Indigenous peoples have accumulated and formed the knowledge system of traditional plant dyeing over a long period of time.

### 3.1.2. The Assessment of Traditional Knowledge on Plant Dyes

The f and CI values have been used as quantitative indicators to assess the application potential of 11 species of dye plants. As shown in Table 5, the f values of *Viburnum cylindricum*, *Buddleja officinalis*, *Strobilanthes cusia*, *Zea mays*, *Oryza sativa*, *Punica granatum*, *Carthamus tinctorius*, *Curcuma longa*, and *Rubia cordifolia* were all over 50%. *V. cylindricum* had the highest f value (80%), followed by *B. officinalis* (75%), as these two plants are the sources of dyes used as raw materials in the national costumes of the Baiyi Indigenous peoples. The f values of *Camellia sinensis* and *Onosma paniculatum* were relatively low. It has also been found that local people often brew *C. sinensis* to make a drink to treat wet sores and ulcers with *O. paniculatum*. The edible value of *C. sinensis* and the medicinal value of *O. paniculatum* is higher than their dyeing value.

**Table 5.** The f value of plant dyes used in the Baiyi community.

| Dye Plant | Use Frequency (f)% |
|---|---|
| *Viburnum cylindricum* | 80 |
| *Buddleja officinalis* | 75 |
| *Rubia cordifolia* | 72 |
| *Zea mays* | 70 |
| *Oryza sativa* | 64 |
| *Strobilanthes cusia* | 60 |
| *Punica granatum* | 55 |
| *Carthamus tinctorius* | 53 |
| *Curcuma longa* | 51 |
| *Camellia sinensis* | 45 |
| *Onosma paniculatum* | 43 |

The cultural importance indices of local dye plants used for dyeing, edible, and medicinal purposes were calculated as shown in Table 6. Except for *O. paniculatum*, the 10 other dye plants are used for dyeing, food, and medicinal applications. The high cultural importance indices of *V. cylindricum*, *B. officinalis*, *R. cordifolia*, and *Z. mays* indicate that the Baiyi Indigenous peoples are familiar with the different usages of these plant dyes. These values also reveal that these plants play important roles in the plant dyeing culture of the Baiyi Indigenous peoples. *V. cylindricum* had the highest CI value (2.20), as it is widely used. Baiyi Indigenous peoples not only use *V. cylindricum* to dye their ethnic costumes, but they also use *V. cylindricum* roots to make a soaking wine as a drink. *O. paniculatum* had the lowest CI value (0.87) because this plant is less commonly used by the local people. Besides, it is mostly used as a medicine and rarely as a dye. Nonetheless, *O. paniculatum* has special significance to some Baiyi Indigenous peoples.

### 3.1.3. The Traditional Knowledge of Plant Dyeing to Make Ethnic Costume

The traditional plant dyeing craft of the Baiyi Indigenous peoples is exemplified by the dyeing of ethnic clothing. The traditional costume of the Baiyi Indigenous peoples is called Huo Cao Yi (Figure 2), which is made of plants. Textiles are made of warp and weft fibers: warp fibers are held stationary in the textile frame, and weft fibers are woven under and over the warp fibers. Warp fibers are made from the stem bark of *Boehmeria nivea* (L.) Gaudich., and weft fibers are made from the fluff on either lower (abaxial) of *Gerbera delavayi* leaves. There are three steps in the dyeing craft of ethnic costumes by the Baiyi Indigenous peoples, as described below.

**Table 6.** Cultural importance indices of dye plants used by the Baiyi Indigenous peoples.

| Species Name | Dye | Usage Food | Medicine | Cultural Importance Index (CI) |
|---|---|---|---|---|
| *Viburnum cylindricum* | 278 | 87 | 270 | 2.20 |
| *Buddleja officinalis* | 265 | 155 | 200 | 2.15 |
| *Rubia cordifolia* | 249 | 88 | 276 | 2.13 |
| *Zea mays* | 207 | 250 | 132 | 2.05 |
| *Oryza sativa* | 171 | 288 | 102 | 1.95 |
| *Strobilanthes cusia* | 180 | 130 | 230 | 1.88 |
| *Punica granatum* | 132 | 265 | 128 | 1.82 |
| *Carthamus tinctorius* | 141 | 96 | 264 | 1.74 |
| *Curcuma longa* | 86 | 156 | 208 | 1.56 |
| *Camellia sinensis* | 65 | 238 | 55 | 1.24 |
| *Onosma paniculatum* | 46 | 0 | 205 | 0.87 |

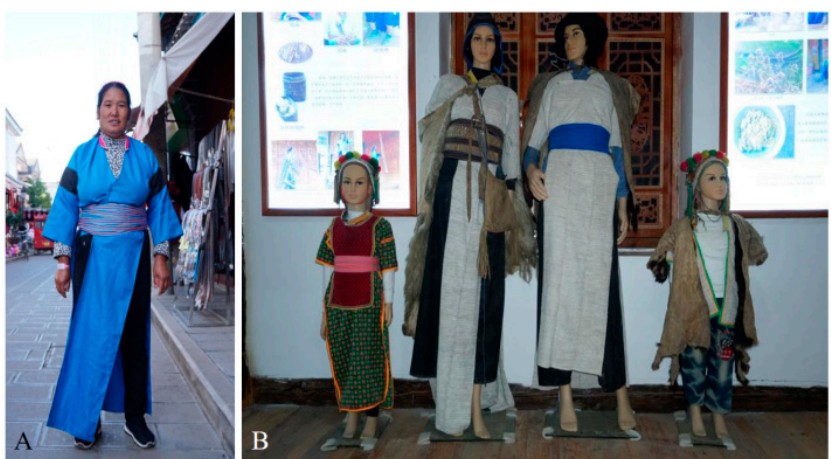

**Figure 2.** The ethnic costume of Baiyi Indigenous peoples. ((**A**) shows the modern ethnic costume of Baiyi women; (**B**) shows the traditional ethnic costume of Baiyi Indigenous peoples).

Step 1: Dyeing Warp Fibers with *Buddleja officinalis*

The flowers of *B. officinalis* are used to dye warp fibers, which are picked in March every year and dried for easy storage by Baiyi Indigenous peoples. These flowers are placed into a stone tub and ground with a wooden mallet, with water added until they turn into a pulp. The materials are boiled in a pot for 30 min and filtered through gauze to produce a dye solution. Meanwhile, the washed warp fibers are boiled in plant ash (raw materials come from *Pinus yunnanensis* Franch.) water with a little lard for one day and one night; according to the Baiyi Indigenous peoples, this step increases the intensity of the coloring. The prepared warp fibers are dyed for half a day in *B. officinalis* dye solution, during which the material is stirred with a wooden board to prevent uneven dyeing (Figure 3). The dyed warp fibers are dried, cleaned with water, dried again.

Step 2: Dyeing Weft Fibers with *Viburnum cylindricum*, *Zea mays*, and *Oryza sativa*

The weft fibers are colored with dye from *V. cylindricum*, *Z. mays*, and *O. sativa*. The Baiyi Indigenous peoples pick and wash *V. cylindricum* leaves, place them in a stone tub and pound them with a wooden stick. During this period, water is added multiple times in small amounts to produce dye pulp from *V. cylindricum*. The weft fibers are dyed with the dye pulp for 2 days, mordanted with black mud for the whole day, then washed and dried for another day (Figure 4). According to the natives, the color of the fabric is determined by the number of repetitions of the dyeing and mordanting steps. The natives believe that black mud functions as a mordant, which fixes the color and enriches the color of the original dye during the dyeing process. The fruit of *Z. mays* is collected by the local people every year in May and June. The fruit of *O. sativa* is placed in water for half a day or 1 day.

Subsequently, *Z. mays* or *O. sativa* fruits are placed in a stone mill and milled with water to produce dyeing pulp. The weft fibers are, respectively, dyed with dyeing pulp of *Z. mays* and *O. sativa* for 1 day and subsequently dried (Figure 4).

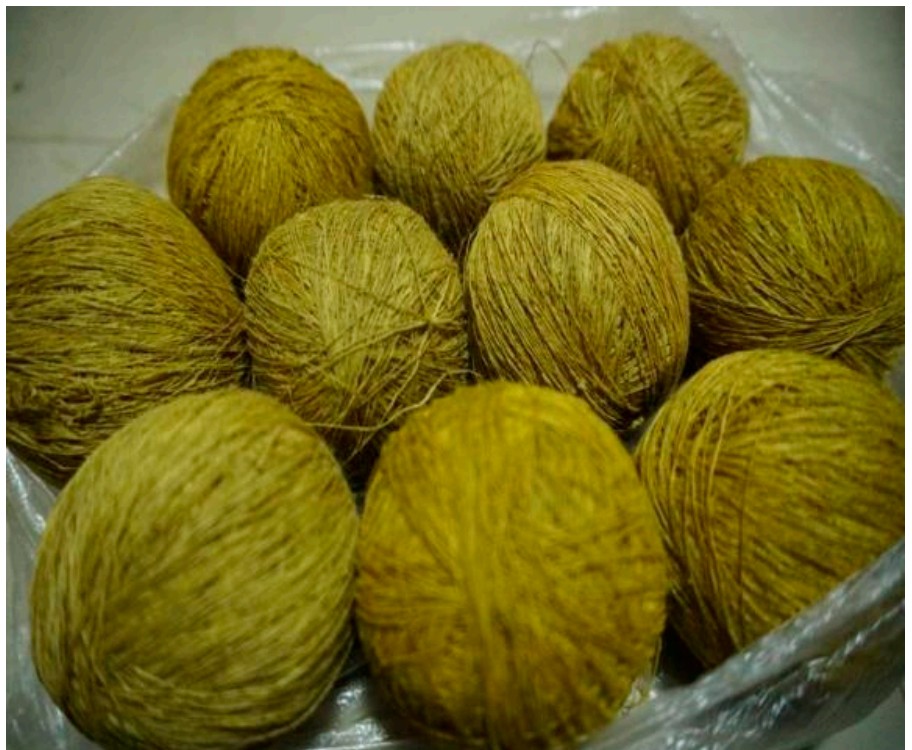

**Figure 3.** The dyed warp fibers with *Buddleja officinalis*.

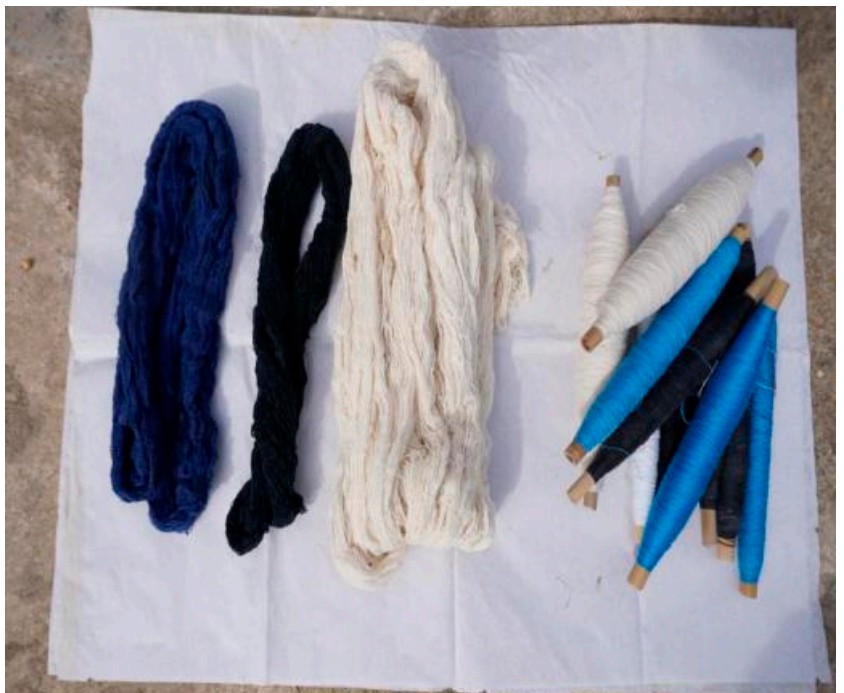

**Figure 4.** The dyed weft fibers with *Viburnum cylindricum*, *Zea mays* and *Oryza sativa*.

Step 3: Weaving the Dyed Warp and Weft Fibers into an Ethnic Costume

The dyed warp and weft fibers are installed and woven in a traditional textile machine. The Huo Cao Yi is made of dark blue, light blue, and yellow fibers alternating with white fibers, and this entire process takes several months to make. Huo Cao Yi has great significance for the Baiyi Indigenous peoples (Figure 5), as it is a costume worn at traditional weddings and funerals. Consequently, this costume serves as a link between generations of Baiyi Indigenous peoples.

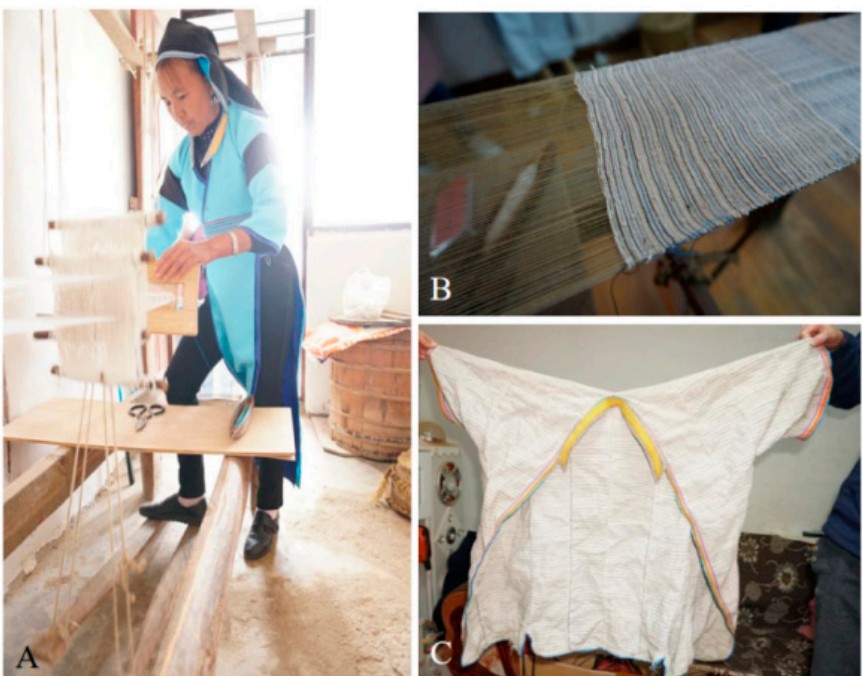

**Figure 5.** The textile craft of traditional ethnic costume of Baiyi Indigenous peoples. ((**A**) shows the weaving process of dyed warp and weft fibers in a spinning machine; (**B**) shows traditional cloth with dyed warp and weft fibers; (**C**) shows sewn traditional ethnic costume).

*3.2. Color Measurements of Fabric Dyed with Viburnum cylindricum*

3.2.1. The Optimum Dyeing Craft Using *Viburnum cylindricum*

Based on field research, it was found that the fibers from the back (abaxial surface) of *G. delavayi* leaves were dyed with *V. cylindricum*, and the fiber structure of this fluff was similar to cotton fibers [43]. Therefore, the experiments of cotton fabric dyed with *V. cylindricum* (Figure 6) were conducted with an orthogonal array design. The $K/S$ and color character values of the nine dyeing experiments are shown in Table 7.

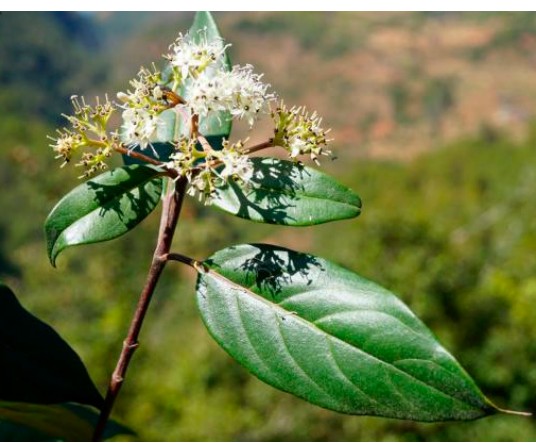

**Figure 6.** The *Viburnum cylindricum* in Baiyi community (Photo taken by Yuying Xie).

**Table 7.** The $K/S$ values and colorimetric data for dyed cotton fabric in the nine dyeing experiments.

| No. | A | B | C | L* | a* | b* | *K/S* |
|---|---|---|---|---|---|---|---|
| 1 | 1 | 1 | 1 | 69.02 | 2.18 | 23.69 | 3.51 |
| 2 | 1 | 2 | 2 | 68.24 | 2.68 | 25.43 | 2.96 |
| 3 | 1 | 3 | 3 | 65.78 | 3.55 | 26.26 | 2.90 |
| 4 | 2 | 1 | 2 | 75.10 | 2.67 | 21.58 | 3.55 |
| 5 | 2 | 2 | 3 | 65.62 | 2.89 | 23.34 | 3.24 |
| 6 | 2 | 3 | 1 | 70.72 | 1.23 | 19.95 | 4.51 |
| 7 | 3 | 1 | 3 | 68.26 | 2.16 | 23.12 | 3.53 |
| 8 | 3 | 2 | 1 | 67.04 | 1.91 | 24.49 | 3.30 |
| 9 | 3 | 3 | 2 | 63.32 | 3.39 | 26.22 | 2.17 |

The results of the range analysis of the R and k values for each factor are listed in Table 8. The $K/S$ value was selected as an indicator to analyze the optimum conditions for dyeing [26]. The influence degrees of the four factors on $K/S$ values were C > A > B. This finding indicates that pH was the most influential factor, followed by temperature, while time was the least influential factor based on the $K/S$ value. The $K/S$ value was optimized under the dyeing conditions of $C_1A_2B_1$ shown in Table 8. Moreover, when the dyeing conditions were $A_2B_1C_1$, $C_2A_1B_3$, and $A_1C_2B_2$, optimum L*, a*, and b* values were obtained, respectively. Consequently, when dyeing was performed at pH = 5 for 30 min at 60 °C, the dyeing craft of *V. cylindricum* dyed cotton fabric was optimized.

**Table 8.** Range analysis of dyed cotton fabric for the orthogonal array with three experimental factors (A, B, and C).

| | | A | B | C |
|---|---|---|---|---|
| $K/S$ | $k_1$ | 3.12 | 3.53 | 3.77 |
| | $k_2$ | 3.77 | 3.17 | 2.89 |
| | $k_3$ | 3.00 | 3.19 | 3.22 |
| | R | 0.77 | 0.36 | 0.88 |
| | Factors order | | C > A > B | |
| | Optimum factors | | $C_1A_2B_1$ | |
| L* | $k_1$ | 67.68 | 70.79 | 68.93 |
| | $k_2$ | 70.48 | 66.97 | 68.89 |
| | $k_3$ | 66.21 | 66.61 | 66.55 |
| | R | 4.27 | 4.19 | 2.37 |
| | Factors order | | A > B > C | |
| | Optimum factors | | $A_2B_1C_1$ | |
| a* | $k_1$ | 2.80 | 2.34 | 1.77 |
| | $k_2$ | 2.26 | 2.49 | 2.91 |
| | $k_3$ | 2.49 | 2.72 | 2.87 |
| | R | 0.54 | 0.39 | 1.14 |
| | Factors order | | C > A > B | |
| | Optimum factors | | $C_2A_1B_3$ | |
| b* | $k_1$ | 25.13 | 22.80 | 22.71 |
| | $k_2$ | 21.62 | 24.42 | 24.41 |
| | $k_3$ | 24.61 | 24.14 | 24.24 |
| | R | 3.50 | 1.62 | 1.70 |
| | Factors order | | A > C > B | |
| | Optimum factors | | $A_1C_2B_2$ | |

### 3.2.2. Comparison of Biomordants and Metallic Mordants

The $K/S$ values of dyed fabrics were enhanced by using biomordants and (to a lesser degree) metallic mordants, as shown in Table 9. The $K/S$ values of dyed fabrics treated with biomordants were higher than those treated with metallic mordants using the same mordanting method. For example, under the simultaneous mordanting method, the $K/S$

values of dyed fabrics with black mud and gum rosin were 5.37 and 8.96, respectively, which is higher than the $K/S$ values of dyed fabrics with KAl(SO$_4$)$_2$ (3.55) and CuSO$_4$ (2.94). The highest $K/S$ value of dyed fabrics was obtained using soybean with the post-mordanting method (10.40), while the lowest value was obtained using SnCl$_2$ with the pre-mordanting method (2.2). The highest $K/S$ values of dyed fabrics were achieved using biomordants. These findings reveal that the effects of biomordants could be better than that of metallic mordants for cotton fabric dyed with *V. cylindricum* with all three mordanting methods. Table 9 shows that for black mud, the $K/S$ values of dyed fabrics were higher using the post-mordanting method compared to the simultaneous and pre-mordanting methods. The result reveals that the traditional method for fabric dyeing with *V. cylindricum* and black mud in post-mordnting is effective with great potential in practical production with convincing evidence. Moreover, it was obvious that the color of *V. cylindricum* dyed cotton fabric was green and brownish black. With the various changes of color of dyed fabric, the color characteristic value (L, a, b value) also varied, which were determined by the different mordants and mordanting methods.

**Table 9.** $K/S$ values and colorimetric data for cotton fabric dyed with *Viburnum cylindricum* using mordants.

| Mordant | Mordanting Method | L | a | b | $K/S$ | Dyed Fabric |
|---|---|---|---|---|---|---|
| None | | 61.51 | 0.70 | 19.48 | 4.61 | |
| KAl(SO$_4$)$_2$ | Pre-mord. | 60.55 | 3.76 | 27.67 | 2.67 | |
| | Simult. mord. | 65.71 | 1.75 | 26.13 | 3.55 | |
| | Post-mord. | 71.39 | 1.22 | 23.06 | 4.84 | |
| CuSO$_4$ | Pre-mord. | 62.05 | 3.51 | 29.07 | 2.24 | |
| | Simult. mord. | 57.79 | 6.68 | 23.61 | 2.94 | |
| | Post-mord. | 56.49 | 6.52 | 23.57 | 3.12 | |
| SnCl$_2$ | Pre-mord. | 64.48 | 5.87 | 35.09 | 2.20 | |
| | Simult. mord. | 73.08 | 1.65 | 29.05 | 4.68 | |
| | Post-mord. | 71.91 | 1.59 | 30.00 | 4.52 | |
| Black mud | Pre-mord. | 64.35 | 6.63 | 28.63 | 5.23 | |
| | Simult. mord. | 51.47 | 20.50 | 13.84 | 5.37 | |
| | Post-mord. | 51.49 | 23.52 | 4.01 | 7.04 | |

**Table 9.** *Cont.*

| Mordant | Mordanting Method | L | a | b | *K/S* | Dyed Fabric |
|---|---|---|---|---|---|---|
| Gum rosin | Pre-mord. | 68.32 | 0.80 | 24.36 | 5.62 | |
| | Simult. mord. | 75.78 | 0.21 | 19.15 | 8.96 | |
| | Post-mord. | 74.16 | 0.74 | 21.59 | 9.39 | |
| Soybean | Pre-mord. | 52.37 | 19.90 | 6.38 | 6.43 | |
| | Simult. mord. | 75.15 | 0.11 | 16.42 | 8.97 | |
| | Post-mord. | 75.54 | -0.26 | 14.82 | 10.40 | |

Note: Pre-mord, pre-mordanting; Simult. mord, simultaneous mordanting; Post-mord, post-mordanting.

The mordanting effect with the same mordanting method of different natural mordants follows a descending order of strength as soybean> gum rosin> black mud which has been illustrated in Table 9. The dyeing mechanism of black mud mainly depends on the reaction of metal ions in the mud with dye molecules and fabric [44]. Soy protein increases the absorption of the dye by forming chemical bonds with dye molecules [22]. By contrast, the main effect of gum rosin is its physical attachment to the fiber surface, followed by cross-linking between the rosin molecules and dye fibers and the intermolecular force of hydrogen bonds [45].

### 3.3. Colorfastness Properties of Fabric Dyed with Viburnum cylindricum

The color fastness values of fabrics dyed with *V. cylindricum* are shown in Table 10. The color staining fastness of dyed fabrics with mordants reached 4–5 and 5, and the color change fastness of dyed fabrics with mordants was 2–3, 3, 3–4, 4, 4–5, and 5, depending on the mordant and mordanting method used. In general, these values are higher than the color fastness of dyed fabrics without mordant. The color fastness of dyed fabrics using biomordants was similar or superior to that using metallic mordants. The color change fastness under washing was enhanced by mordanting with black mud and gum rosin, with scores of 3–4, compared to only 2–3 using stannous chloride with the pre-mordanting method.

**Table 10.** Colorfastness of cotton fabrics dyed with *Viburnum cylindricum*.

| Mordant | Mordanting Method | Washing Fastness | | Rubbing Fastness | | | | Perspiration Fastness | | | |
|---|---|---|---|---|---|---|---|---|---|---|---|
| | | | | Dry | | Wet | | Acidic | | Alkaline | |
| | | CC | CS | CC | CS | CC | CS | CC | CS | CC | CS |
| None | | 2 | 4–5 | 4 | 5 | 4 | 3 | 4 | 4 | 2–3 | 3 |
| Black mud | Pre-mord. | 3–4 | 5 | 4–5 | 5 | 4 | 4–5 | 4 | 5 | 4 | 4–5 |
| | Simult. mord. | 2–3 | 5 | 4–5 | 5 | 4 | 5 | 4 | 5 | 3–4 | 5 |
| | Post-mord. | 3–4 | 5 | 4 | 5 | 4 | 5 | 4 | 5 | 3 | 4–5 |
| Gum rosin | Pre-mord. | 3–4 | 5 | 4 | 5 | 4–5 | 4–5 | 4–5 | 5 | 4–5 | 4–5 |
| | Simult. mord. | 3 | 5 | 4 | 5 | 4–5 | 5 | 4–5 | 5 | 3–4 | 5 |
| | Post-mord. | 2–3 | 5 | 4–5 | 5 | 4–5 | 5 | 4 | 5 | 3–4 | 5 |

**Table 10.** *Cont.*

| Mordant | Mordanting Method | Washing Fastness | | Rubbing Fastness | | | | Perspiration Fastness | | | |
|---|---|---|---|---|---|---|---|---|---|---|---|
| | | | | Dry | | Wet | | Acidic | | Alkaline | |
| | | CC | CS | CC | CS | CC | CS | CC | CS | CC | CS |
| Soybean | Pre-mord. | 3 | 5 | 4 | 5 | 4 | 4–5 | 4 | 5 | 3–4 | 4–5 |
| | Simult. mord. | 2–3 | 5 | 4–5 | 5 | 4 | 4–5 | 4–5 | 5 | 4 | 4–5 |
| | Post-mord. | 4 | 5 | 4 | 5 | 4 | 4–5 | 5 | 5 | 4–5 | 4–5 |
| $KAl(SO_4)_2$ | Pre-mord. | 3 | 5 | 4 | 5 | 4 | 4–5 | 4 | 5 | 4 | 4–5 |
| | Simult. mord. | 3–4 | 5 | 4–5 | 5 | 4–5 | 4–5 | 4–5 | 5 | 4–5 | 5 |
| | Post-mord. | 3 | 5 | 4–5 | 5 | 4–5 | 5 | 4 | 5 | 4 | 5 |
| $SnCl_2$ | Pre-mord. | 2–3 | 5 | 4 | 5 | 4 | 4–5 | 4–5 | 5 | 4 | 4–5 |
| | Simult. mord. | 2–3 | 5 | 4–5 | 5 | 4 | 5 | 4–5 | 5 | 4 | 5 |
| | Post-mord. | 2–3 | 5 | 4 | 5 | 4 | 4–5 | 4 | 5 | 4 | 5 |
| $CuSO_4$ | Pre-mord. | 3 | 5 | 5 | 5 | 4 | 4–5 | 4 | 4–5 | 3–4 | 4–5 |
| | Simult. mord. | 2–3 | 5 | 4–5 | 5 | 4 | 5 | 4 | 5 | 4 | 4–5 |
| | Post-mord. | 2–3 | 5 | 4 | 5 | 4–5 | 5 | 4 | 5 | 3–4 | 5 |

Notes: CC: Color change, CS: Color staining, 1: very poor, 2: poor, 3: moderate, 4: good, 5: excellent.

## 4. Discussion

### 4.1. The Application Potential of the Traditional Plant Dyeing Craft of the Baiyi Indigenous Peoples

Various plant dyes are used by ethnic minorities for dyeing, which are becoming increasingly favored due to their properties of non-toxic, insect-proof, sterile, anti-UV and other functional features [46–48].

An analysis of use frequency (f) and cultural significance (CI) revealed that the f and CI values of *V. cylindricum* and *B. officinalis* were highest among plants. *V. cylindricum* and *B. officinalis* are the materials of dyes that are applied to color fabrics used for ethnic costumes of the Baiyi Indigenous peoples, with the mordant of black mud and plant ash water with lard and the method of post-mordanting. Remarkably, Yi people use mud to dye Chalva costumes in Liangshan using the pre-mordanting method [49]. Mud is also used for dyeing to make the traditional textile Bogolan (or mud cloth) by indigenous people of the Bamana Tribe of Mali [44]. Local people use *Anogeissus leiocarpa* to dye cloth, and mud is used to draw patterns in impregnated cloth with plant dyes [44]. Similarly, mud and ash are used as mordants in traditional plant dyeing by the Tai-Lao ethnic group in northeastern Thailand [50]. However, the use of *V. cylindricum* as a dye plant has not been reported. Thus, the traditional plant dyeing craft of Baiyi is unique.

The 11 species of plants that are used to make dye have various medicinal and other properties in Table 4. *V. cylindricum* serves as a raw material for traditional medicine in the Dai community and the roots, bark, leaves, and flowers of which can be used to clear heat and detoxify, clean abscesses, and treat cough, dysentery, rheumatic pain, and injuries [51]. *B. officinalis* is not only used to dye rice by the ethnic community in Yunnan [26,31], but it can also clear heat, dispel "fire," nourish the liver, and brighten the eyes [52]. *R. cordifolia* is a traditional dye plant worldwide that is used in different crafts in various regions [53–55]. This plant is boiled in water to extract pigment by the Baiyi Indigenous peoples. It also has the medicinal effect of cooling blood to remove stasis and hemostasis, especially in gynecological diseases [56].

### 4.2. The Current State of Plant Dyeing and Related Traditional Knowledge in the Baiyi Community

We interviewed 288 key informants in this study. This group included more women than men and more older informants than young people. We determined that men extract the dye, while women are responsible for the collection of plant dyes, dyeing, weaving clothes, and designing patterns. Contrarily, the men hardly join in the whole process of plant dyeing, and the traditional knowledge of plant dye is dominated by women in the Baiku Yao community [33]. Considering that the Baiyi Indigenous peoples do not have

their own written language, the traditional knowledge of plant dyeing is passed down orally. The Baiyi costume is mostly worn by locals during festivals, weddings, and funerals.

With the development of society and the influence of foreign culture, Baiyi costumes are worn less frequently by young people. Besides, many young people do not believe that traditional plant dyeing is profitable. Because traditional dyeing methods are complicated and laborious, young people do not want to perform traditional plant dyeing. Moreover, due to the increased population and the destruction of mountain forests in recent years, the availability of plant dye resources has gradually decreased. Furthermore, young people do not generally appreciate the value of protecting and cultivating dye plants, and their knowledge of traditional plant dyes is limited. These factors are leading to the gradual disappearance of local traditional knowledge on the sustainable use and management of plant dyes. And this situation is similar to the current situation of the traditional knowledge of plant dyeing of the Bai ethnic group, but the traditional knowledge of plant dyeing is mainly affected by economic development in the Bai ethnic community [31].

*4.3. The Protection and Inheritance of Traditional Plant Dyeing by the Baiyi Indigenous Peoples*

Through field investigation and experimental verification, we found that the color fastness of traditional plant dyeing is low, the dye plant resources are limited, the cost of plant dyeing is high, the process of traditional plant dyeing is complicated, and traditional knowledge of plant dyeing is gradually disappearing in the Baiyi community.

It is important to protect and maintain the traditional plant dyeing craft of the Baiyi Indigenous peoples. The following suggestions could help accomplish this goal: (1) The existing dye plants and their seeds should be collected to preserve the germplasm resources of dye plants and study the cultivation of these plants. (2) The current traditional knowledge of plant dyeing should be documented, collected, and studied systematically through ethnobotanical investigations to develop the relevant data archives. (3) A community market should be established for traditional plant dyeing products and it is necessary to create sufficient profits to stimulate local people to pursue the production and sale of traditional plant dyes and products. (4) The properties of traditional plant dyes should be verified and optimized, which is a necessary step for protecting and promoting traditional dyeing using plants in the Baiyi community. (5) Local governments should value traditional plant dyeing, develop policies, and provide economic support for the plant dyeing industry and the study of plant dyeing. (6) Market research should be performed to exploit new uses of traditional plant dyeing to expand its application range.

**5. Conclusions**

The ethnobotanical investigation of plant dyeing by the Baiyi Indigenous peoples has been conducted among four villages of Yi Nationality Township in Liuhe, Heqing County, Yunnan Province, China. Eleven species of plants belonging to 10 families and 11 genera are used as dyes by the Baiyi community. These dye plants also have medicinal properties and a wide range of other uses. Moreover, Baiyi Indigenous peoples have widespread and unique traditional plant dyeing knowledge and dyeing technology. Similarly, the number of dye plants and the traditional knowledge of related plant dyeing were documented and studied through ethnobotanical investigations in Dong and Bai communities [31,32]. The craft of traditional plant dyeing by Dong and Bai ethnicities still is not optimized by researchers. In this study, *V. cylindricum* was selected as a source of plant dye for an orthogonal array experiment in which we quantitatively analyzed f and CI values to verify and optimize the properties of traditional plant dyeing. The optimized dyeing conditions for *V. cylindricum* were pH = 5, 30 min, and 60 °C. The effects of biomordants could be compared with the effects of metallic mordants for improving the dyeing properties of *V. cylindricum* dyed cotton fabrics.

This study examined plant dyeing and related traditional knowledge of the Baiyi Indigenous peoples in Heqing and confirmed the potential of traditional plant dyeing. More attention should be paid to the research and development of traditional plant dyeing

technology to help make plant dyes become sustainable substitutes for synthetic dyes. Given the impact of globalization and rapid social development on local communities in China, it is essential to explore plant dye resources and related traditional knowledge to protect and sustainably use these resources. Such information will help promote the economic development of local communities, maintain ethnic and cultural inheritance, and conserve biodiversity.

**Author Contributions:** Conceptualization, L.Y. and S.P.; methodology, A.I. and R.Y.; validation and formal analysis, R.Y. and X.Y.; investigation and visualization, R.Y., Y.X. and L.Y.; writing—original draft preparation, R.Y.; writing-review, and editing, A.I., Y.X. and L.Y.; supervision, project administration and funding acquisition, L.Y. All authors have read and agreed to the published version of the manuscript.

**Funding:** The fabrics used in this study were provided by Esquel Enterprises Ltd. This study was supported by the National Nature Science Foundation of China (No. 31670340 and 31970357), Strategic Priority Research Program of Chinese Academy of Sciences (No. XDA20050204, XDA19050301, and XDA19050303), and the Second Monpa Plateau Scientific Expedition and Research Program (No. 2019QZKK0502).

**Institutional Review Board Statement:** Not applicable.

**Data Availability Statement:** Experimental data are available from the corresponding author upon reasonable written request.

**Acknowledgments:** We are very thankful to the inheritors of Baiyi culture and local people in Liuhe Yi Nationality Township, Heqing County, Dali, Yunnan, China, who have shared valuable information related to traditional knowledge and plant dye resources. We greatly appreciate their cooperation and hospitality. We are also grateful to Chiang Mai University for partial support in this study.

**Conflicts of Interest:** The authors declare that they have no competing interests.

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
