# Peer review of "Ethnobotanical Research on Dye Plants Used by the Baiyi Indigenous Peoples’ from Heqing County, Dali, Yunnan, China"

_diversity, doi:10.3390/d15070856_

Round 1

Reviewer 1 Report

I had the opportunity to review the paper entitled "Ethnobotanical research on dye plants used by the Baiyi people from Heqing County, Dali, Yunnan, China". Although ethnobotany as a subject is not strictly within the scope of the journal, the manuscript makes some important recommendations about diversity preservation, it can therefore be considered for publication if it meets the other criteria. I have added my comments directly on the PDF copy that I am uploading together with this review, however, I would like to highlight a few points here.

1. The manuscript makes for an interesting read and on the whole is well-written. Nonetheless, there are certain statements which are difficult to follow due to sentence misconstructions and grammatical errors. In addition, some of the sentences are incomplete. For example, the very first sentence in the Abstract reads "with the advantage of eco-friendly, plant dyes have been noticed...", does not make sense.

2. I was a bit concerned about there was no written informed consent, however, I believe that verbal informed consent is also acceptable. But perhaps an explanation of why only verbal informed consent was considered is in order. 

3. I cannot follow the system for the arrangement of the plants listed in Table 4, it seems to be random, consider arranging alphabetically. In addition, the scientific names of plants should be accompanied by author citations. 

4. It is not clear in which herbarium the voucher specimens are housed, the voucher numbers for plants used in the research should also be included.

I have already commented about the quality of the English Language above, only moderate language editing is required.  

Author Response

For reviewer 1: 

Comments and Suggestions for authors

I had the opportunity to review the paper entitled "Ethnobotanical research on dye plants used by the Baiyi people from Heqing County, Dali, Yunnan, China". Although ethnobotany as a subject is not strictly within the scope of the journal, the manuscript makes some important recommendations about diversity preservation, it can therefore be considered for publication if it meets the other criteria. I have added my comments directly on the PDF copy that I am uploading together with this review, however, I would like to highlight a few points here.

Comment 1.1: The manuscript makes for an interesting read and on the whole is well-written. Nonetheless, there are certain statements which are difficult to follow due to sentence misconstructions and grammatical errors. In addition, some of the sentences are incomplete. For example, the very first sentence in the Abstract reads "with the advantage of eco-friendly, plant dyes have been noticed...", does not make sense.

Response 1.1: Yes, we agree with your suggestion. Thank you for your kind comment, we have revised this part in terms of your comment. We have revised the marked sentences in the manuscript.

Comment 1.2: I was a bit concerned about there was no written informed consent, however, I believe that verbal informed consent is also acceptable. But perhaps an explanation of why only verbal informed consent was considered is in order.

Response 1.2: Yes, we are in favor of your comment. Thank you for your assessment! The verbal informed consent was used because many Baiyi people could not read Chinese characters, and many Baiyi people have a low level of education.

Comment 1.3: I cannot follow the system for the arrangement of the plants listed in Table 4, it seems to be random, consider arranging alphabetically. In addition, the scientific names of plants should be accompanied by author citations.

Response 1.3: Yes, we agree with your suggestion. Thank you for your suggestion. We have arranged the plants listed in Table 4 considering arranging alphabetically. And we have also added author citations in the scientific names of plants.

Comment 1.4: It is not clear in which herbarium the voucher specimens are housed, the voucher numbers for plants used in the research should also be included.

Response 1.4: Yes, we agree your advice. Thank you for your advice. We have supplemented this part in section of “2.2. Field research” in line 131 and line 135. The local plant dyes were identified, classified and collected as document specimens and samples to form the ethnobotanical catalog of plant dyes of the Baiyi people. The scientific names of dye plants were classified and confirmed through APG IV. And plant specimens were reserved at Herbarium, Kunming Institute of Botany, CAS.

Reviewer 2 Report

Comments and Suggestions for authors

The manuscript is well articulated. Based on ethnobotanical research, the authors have explored plant species used as dye by the Baiyi indigenous peoples’ from Heqing, Dali, Yunnan, China. My comments and suggestions are as follows:

·         Uniformity: In the manuscript, the authors have used Baiyi ethnic groups differently as “Indigenous communities”; “Ethnic communities”; “Local communities”; “Baiyi people”. Also it seems that Baiyi ethnic groups consists of more than one subgroup. So, a widely accepted terminology used by United Nations agreements is “Indigenous peoples’ and local communities (IPLCs)”. Therefore, I suggest to use the well accepted terminology for Baiyi ethnic groups, and that is Baiyi Indigenous peoples’ and local communities.  

·         Scientific name of plant species: I suggest to use latest nomenclature of the plant taxa using APG IV. For example (Table 4) Viburnum cylindricum is kept under Viburnanceae (NOT Caprifoliaceae); Buddleja officinalis under Scrophulariaceae (NOT Loganiaceae); Punica granatum under Lythraceae (NOT Punicaceae); Carthamus tinctorius under Asteraceae. Also, make correction of Camelilia sinensis Kitamura Kitamura (Table 4, NP 13).  

·         Map resolution: Check quality of the map – Figure 1.  

·         Justify the Table 4: I suggest to bring the table in a clear format.

·         English Language: Check the manuscript thoroughly. Some sentences/lines to be rechecked and corrected include – Lines 16-17; Lines 38-39; Lines 71-72; Lines 155-156; Lines 211-212; Line 240; etc.  

·         Plant ash: Line 265, better to mention the plant species used as ash, if you knowS. There are some publications mentioning the use of Oak species ash in Nepal during the process of dyeing.

·         Caption: Revisit the caption of Figure 6.

·         Scientific name: May I suggest to write scientific name of all plant species including those mentioned in the “References” part in italics. Also suggested to revisit the scientific name. 

·         Thorough Manuscript (MS) check: I strongly recommend the authors to read carefully the MS, and pay an attention to the spelling mistakes, use capital letters when you provide names of village or province, etc.

Moderate editing of English Language required

Author Response

For reviewer 2: 

Comments and Suggestions for authors

The manuscript is well articulated. Based on ethnobotanical research, the authors have explored plant species used as dye by the Baiyi indigenous peoples’ from Heqing, Dali, Yunnan, China. My comments and suggestions are as follows:

Comment 2.1: Uniformity: In the manuscript, the authors have used Baiyi ethnic groups differently as “Indigenous communities”; “Ethnic communities”; “Local communities”; “Baiyi people”. Also it seems that Baiyi ethnic groups consists of more than one subgroup. So, a widely accepted terminology used by United Nations agreements is “Indigenous peoples’ and local communities (IPLCs)”. Therefore, I suggest to use the well accepted terminology for Baiyi ethnic groups, and that is Baiyi Indigenous peoples’ and local communities.

Response 2.1: Thank you for your comment, and we appreciate this point. And we have replaced “Baiyi people” with “Baiyi Indigenous peoples” in manuscript.

Comment 2.2: Scientific name of plant species: I suggest to use latest nomenclature of the plant taxa using APG IV. For example (Table 4) Viburnum cylindricum is kept under Viburnanceae (NOT Caprifoliaceae); Buddleja officinalis under Scrophulariaceae (NOT Loganiaceae); Punica granatum under Lythraceae (NOT Punicaceae); Carthamus tinctorius under Asteraceae. Also, make correction of Camelilia sinensis Kitamura Kitamura (Table 4, NP 13).

Response 2.2: Thank you for your comment, and we appreciate this point. And we have revised according your suggestion.

Comment 2.3: Map resolution: Check quality of the map – Figure 1.

Response 2.3: Yes, we are in favor of your comment. We have revised Figure 1 for clearer viewing by the reader.

Comment 2.4: Justify the Table 4: I suggest to bring the table in a clear format.

Response 2.4: Yes, we agree with your suggestion. Thank you for your suggestion. We have arranged the plants listed in Table 4 considering arranging alphabetically. And we have also added author citations in the scientific names of plants.

Comment 2.5: English Language: Check the manuscript thoroughly. Some sentences/lines to be rechecked and corrected include – Lines 16-17; Lines 38-39; Lines 71-72; Lines 155-156; Lines 211-212; Line 240; etc.

Response 2.5: Thank you for your comment, and we revised in lines 18-19; lines 42-43; lines 76-77; lines 163; lines 219-221; line 250-252; etc.

Comment 2.6: Plant ash: Line 265, better to mention the plant species used as ash, if you knowS. There are some publications mentioning the use of Oak species ash in Nepal during the process of dyeing.

Response 2.6: Yes, we agree your advice. Thank you for your advice. We have supplemented this part in line 277-278. The raw materials of plant ash come from Pinus yunnanensis Franch in manuscript.

Comment 2.7: Caption: Revisit the caption of Figure 6.

Response 2.7: Yes, we are in favor of your comment. We have picked a new picture in Figure 6.

Comment 2.8: Scientific name: May I suggest to write scientific name of all plant species including those mentioned in the “References” part in italics. Also suggested to revisit the scientific name.

Response 2.8: Yes, we agree your advice. Thank you for your advice. We have amended scientific name of all plant species by italics.

Comment 2.9: Thorough Manuscript (MS) check: I strongly recommend the authors to read carefully the MS, and pay an attention to the spelling mistakes, use capital letters when you provide names of village or province, etc.

Response 2.9: Thank you for your suggestion, and we appreciate this point. And we have carefully checked the manuscript and revised it.

Reviewer 3 Report

Dear authors,

Congratulations on the search. I just suggest refining the results and discussion and revising the citation of references.

Author Response

For reviewer 3: 

Comments and Suggestions for Authors

Dear authors,

Comment 3.1: Congratulations on the search. I just suggest refining the results and discussion and revising the citation of references.

Response 3.1: Sincerely appreciate for your greatly assessment to our work. And we have deleted suggestion (7)-(8) in the section of “4.3. The protection and inheritance of traditional plant dyeing by the Baiyi Indigenous peoples’ ”, meanwhile we have also deleted some sentenced in results.
